# Reprocessing filtering facepiece respirators in primary care using medical autoclave: prospective, bench-to-bedside, single-centre study

Ralf E Harskamp [ORCID],[1] Bart van Straten,[2,3] Jonathan Bouman,[1] Bernadette van Maltha-van Santvoort,[4] John J van den Dobbelsteen,[2] Joost RM van der Sijp,[5,6] Tim Horeman[2]

REH and BvS contributed equally.

For numbered affiliations see end of article.

**Correspondence to**
Dr Ralf E Harskamp;
r.e.harskamp@gmail.com

## ABSTRACT

**Objective** There are widespread shortages of personal protective equipment as a result of the COVID-19 pandemic. Reprocessing filtering facepiece particle (FFP)-type respirators may provide an alternative solution in keeping healthcare professionals safe.

**Design** Prospective, bench-to-bedside.

**Setting** A primary care-based study using FFP-2 respirators without exhalation valve (3M Aura 1862+ (20 samples), Maco Pharma ZZM002 (14 samples)), FFP-2 respirators with valve (3M Aura 9322+ (six samples) and San Huei 2920V (16 samples)) and valved FFP type 3 respirators (Safe Worker 1016 (10 samples)).

**Interventions** All masks were reprocessed using a medical autoclave (17 min at 121°C with 34 min total cycle time) and subsequently tested up to three times whether these respirators retained their integrity (seal check and pressure drop) and ability to filter small particles (0.3–5.0 µm) in the laboratory using a particle penetration test.

**Results** We tested 33 respirators and 66 samples for filter capacity. All FFP-2 respirators retained their shape, whereas half of the decontaminated FFP-3 respirators showed deformities and failed the seal check. The filtering capacity of the 3M Aura 1862 was best retained after one, two and three decontamination cycles (0.3 µm: 99.3%±0.3% (new) vs 97.0±1.3, 94.2±1.3% or 94.4±1.6; p<0.001). Of the other FFP-2 respirators, the San Huei 2920 V had 95.5%±0.7% at baseline vs 92.3%±1.7% vs 90.0±0.7 after one-time and two-time decontaminations, respectively (p<0.001). The tested FFP-3 respirator (Safe Worker 1016) had a filter capacity of 96.5%±0.7% at baseline and 60.3%±5.7% after one-time decontamination (p<0.001). Breathing and pressure resistance tests indicated no relevant pressure changes between respirators that were used once, twice or thrice.

**Conclusion** This small single-centre study shows that selected FFP-2 respirators may be reprocessed for use in primary care, as the tested masks retain their shape, ability to retain particles and breathing comfort after decontamination using a medical autoclave.

## Strengths and limitations of this study

► Pragmatic use of tabletop autoclave to decontaminate and reuse FFP-respirators.
► Combining clinical and laboratory findings to evaluate the safety in terms of shape, ability to retain particles and breathing comfort.
► The study is limited in sample size and restricted to selected FFP-2 and FFP-3 respirators.
► The study is a first of its kind in primary care settings and thus non-validated.
► The study does not provide 'hard' clinical evidence in terms of a randomised trial (ie, reprocessed mask vs usual care).

## INTRODUCTION

General practitioners (GPs) are often the first to evaluate patients with (suspected) COVID-19. This is particularly true in countries where GPs have a gatekeeping role. Given the risk of person-to-person spread, this necessitates the need to wear personal protective equipment.[1 2] Unfortunately, most healthcare facilities are running dangerously low on this equipment.[1 2] In the USA, these critical shortages have resulted in downgrading from respirators to surgical masks and sometimes even resort to home-made cloth masks.[1] Access to adequate supplies is crucial to preventing transmission of pathogens, especially in resource-limited settings.[3] Reports across several countries found that healthcare workers are more at risk of catching severe acute respiratory syndrome coronavirus 2 (SARS-CoV-2) as well as at higher risk of severe COVID-19, possibly due to exposure to higher viral load.[4] The outbreak of COVID-19 in Italy showed that inadequate access to protective equipment is one of the reasons why healthcare workers, and particularly GPs, experienced high rates of infection.[2] Aside from the direct health effects, absenteeism from illness may also negatively affect the health system's capacity

to adequately respond to the COVID-19 pandemic. Moreover, it makes healthcare workers feel unsafe and unprotected, which undermines morale as shown in a report on England's National Health Service health workers experiences.[5]

One of the possible (short-term pragmatic) solutions could be the reuse of equipment, and in particular that of respirators. To reuse a mask or respirator, it should be decontaminated first. The method applied should (1) kill the SARS-CoV-2 virus (diminish the viral load) and (2) keep the mask's protective properties (largely) intact, in terms of filter and fit. In primary care, the medical autoclave is normally used to decontaminate surgical instruments. The process of pressurised moist heat destroys microorganisms by the irreversible coagulation and denaturation of enzymes and structural proteins, and has been shown to be effective in respirators contaminated with other viruses, such as H1N1 influenza virus.[2 6 7] However, the question is whether the respirator's protective properties in terms of filter function and fit will remain intact when exposing the respirator to steam. We therefore set out to study whether the process of steam sterilisation negatively affects the protective properties of commonly used respirators, which are designed to protect the wearer against the inhalation of both droplets and particles suspended in the air.

## METHODS
We reported our findings according to the Strengthening the Reporting of Observational Studies in Epidemiology checklist and guide (online supplementary file l), as well as the general principles of reporting a study using the directions provided by the journal.[8]

### Study design and setting
The study involved the evaluation of available filtering respirators used to evaluate patients with suspected COVID-19 in the Holendrecht Medical Centre, in Amsterdam, the Netherlands. For high-risk patients, the centre provides GPs with filtering facepiece particle (FFP) type 2 or type 3 for personal protection. For the current study, worn respirators were used for reprocessing using a medical autoclave. After the autoclave procedure, the respirators were visually inspected for deformity by two clinical investigators, followed by a seal check. The masks were subsequently marked and sent by courier to the GreenCycl testing laboratory in Utrecht, the Netherlands. At this facility, the decontaminated respirators were tested by two laboratory scientists for their filter capacity. For comparison, the results ware compared with the filter capacity of unused, brand-new respirators that were used as a reference. Moreover, a pressure drop test was performed to evaluate whether the breathing resistance altered by the process of decontamination.

### Decontamination process
The masks were decontaminated for multiple cycles using a cylindrical chamber tabletop autoclave (Kronos S18, release: E.5.47a, Newmed, Quattro Castella, Italy). This type of vacuum autoclave is typically designed for GP and dental practices and has preprogrammed cycles. The size as well as the cycle times differ from autoclaves typically used in hospitals, which are larger and have longer cycle times, but with comparable peak times regarding decontamination. The Kronos S18 autoclave holds a capacity of 18 litres or four respirators. The autoclave has specific programmes for 'solid made of rubber and delicate solids', which includes respirators. The sterilisation programme we used involved a 34 min cycle, of which the first 12 min of the cycle involved preheating, followed by 17 min steam decontamination at a temperature of 121°C, and finished with a 5 min drying process.

### Visual inspection, breathing resistance and user seal check
After decontamination, the respirators were checked for visual deformities of the mask, as well as the elastic straps. Subsequently, the respirators were put on to evaluate whether breathing felt normal, followed by the performance of a user seal check. A negative pressure user seal check was used for all respirators in which the clinical investigator inhaled sharply while blocking the paths for air to enter the facepiece. A successful check is when the facepiece collapsed slightly under the negative pressure that was created with this manoeuvre. For respirators without an exhalation valve, the investigator also performed a positive pressure check by exhaling gently while blocking the paths for air to exit the facepiece. A successful check is when the facepiece was slightly pressurised before increased pressure causes outward leakage.[9]

### Particle penetration test
At the testing laboratory, two independent researchers from the Delft University of Technology tested the masks using a dry particle penetration test set-up (figure 1).[10] The equipment involved a SOLAIR 3100 particle counter (Lighthouse Worldwide Solutions, Fremont, California, USA). The particles are counted within the machine via a tube that is connected to a particle chamber to which the respirator is secured. The transparent lid presses the mask such that it prevents material buckling and creates an airtight seal that only allows air to pass through the material. Before each measurement, a benchmark test is conducted with 28 litres of surrounding air that is sucked through the particle chamber into the particle counter (figure 1, top half). The particle counter measures the particles that are naturally present in the air. During benchmark testing, no mask is installed. During the test measurement, a mask is installed on the particle chamber (figure 1, bottom half). Therefore, the 28 litres of surrounding air is sucked through the filter material of the mask, and the remaining particles are counted in the categories of 0.3, 0.5, 1.0 and 5.0 microns. The

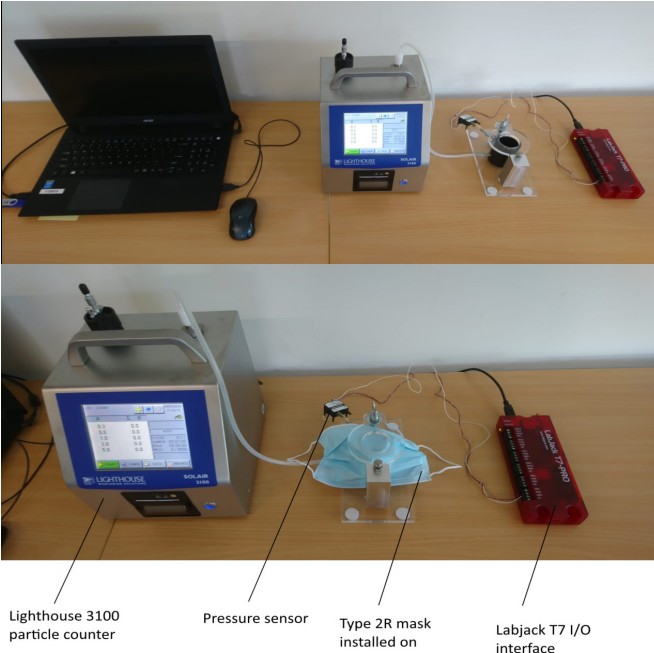

Lighthouse 3100 particle counter

Pressure sensor

Type 2R mask installed on particle chamber

Labjack T7 I/O interface

**Figure 1** Illustration of the measurement set-up used to measure the particle penetration capacity of different respirators. The particle chamber is connected to the Lighthouse 3100 with a custom connecter and 5 mm tube. An adjustable removable transparent lid is used to press the filter material of a mask airtight on the rim of the open particle chamber for accurate measurements.

measurements are compared and the filtering capacity is derived based on the difference in the readings compared with the benchmark test. A lower number of particles counted after filtering in relation to the benchmark test would indicate better filtering performance.[11] The system set-up is more conservative than the NEN-149 standard, which means the resulting filter capacity percentages cannot be translated directly to the known FFP-1, FFP-2 and FFP-3 standards. However, as the filter capacity of a new mask is known, the measurement results do show the remaining filter capacity and therefore indicates whether a mask type deteriorates after steam decontamination.

### Flow resistance

The set-up was expanded with an additional pressure sensor (SDP 816–500 Pa Sensirion #1230319) and flow adjustment valve in order to investigate whether the pressure delta over the mask material changed after decontamination cycles. The Sensirion pressure sensor was connected with a T-piece between the particle chamber and Lighthouse 3100. An additional valve was used to adjust the input pressure within range of the sensor. A LabJack T7 analogue input device was used to convert the output from the pressure sensor to an output voltage of 0–5 V. An output value of 5 V was representing 500 Pa and set as 100% of input pressure. The atmospheric input pressure of 2.42 V was defined as 0% output. Measurements were conducted with a constant air speed of 20.7 m/s at the opening of the particle chamber.

### Outcomes of interest

The outcomes of interest involved (1) signs of deformity of the respirator, which was performed by visual inspection; and (2) the percentage of filtered particles with a diameter of 0.3 µm. This diameter is clinically relevant, given that to meet the FFP-2 standards, a mask should filter 94% of all 0.3 µm particles, whereas 99% of these particles should be filtered to meet the FFP-3 standard.

### Statistical analysis

The study involved descriptive analyses, with numbers and percentages, and comparisons were performed using an alpha of 0.05 for statistical significance. The findings of the filter tests are visually displayed in box plots and presented as mean and SD. We used JASP statistical software V.0.10.2 (University of Amsterdam, the Netherlands).

### Patient and public involvement

Patients and/or the public were not involved in the design, conduct, reporting, or dissemination plans of our research.

## RESULTS

We obtained 33 respirators, of which 28 were used during consultation or high-risk home visits of patients suspected to have COVID-19 at the Holendrecht Medical Centre in March/April 2020. The face masks were FFP-2 respirators (3M Aura 1862+, Maco Pharma ZZM002), FFP-2 respirators with exhalation valve (3M Aura 9322+ and San Huei 2920V) or FFP-3 respirators (Safe Worker 1016). The 28 used respirators (including 4 FFP-3 respirators) underwent decontamination, with the remaining 5 serving as a reference (as they did not to undergo decontamination).

### Visual inspection, breathing resistance and user seal check

After the decontamination process, all FFP-2 respirators retained their shape and were without visible damage. When fitting, the elastic bands of all masks still functioned normally, with no difference from non-decontaminated masks in terms of breathing resistance. The seal checks also did not reveal significant air leakage suggesting poor fit. However, unlike the FFP-2 respirators, two out of the four FFP-3 respirators (50%) showed signs of deformation, with a crumbled appearance, and failed seal check test.

### Filter capacity of decontaminated respirators

For the particle penetration test, a total of 66 samples were tested from 33 respirators. The results of the filter capacity for 0.3 microns are illustrated in figure 2 and for larger particles are displayed in table 1. Of the tested FFP-2 respirators, we found that the 3M Aura 1862+ remained close to its original filtering capacity after one-time, two-time and three-time decontamination (0.3 µm: 99.3%±0.3% vs 97.0±1.3, 94.2±1.3% or 94.4±1.6, respectively, p<0.001). The 3M Aura 9322+ (with valve) had a filter capacity of 96.8%±0.2% without decontamination vs 91.0%±1.4% and 77.5%±2.1% after one-time or two-time

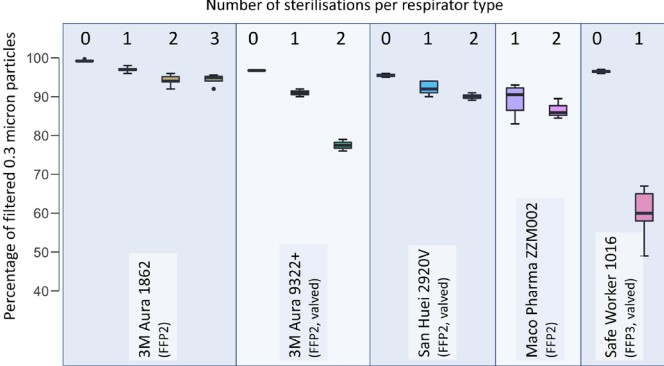

**Figure 2** Filter quality of autoclave-decontaminated respirators (retainment of 0.3 µm particles) of unused and one-time, two-time, and three-time autoclave sterilisation.

decontamination (p<0.001). The Maco Pharma ZZM002 FFP-2 mask did not have a reference mask, but after one-time and two-time decontamination, the filter capacities were 89.3%±3.9% and 86.6%±2.6%, respectively. The San Huei 2920V respirator had 95.5%±0.7% at baseline vs 92.3%±1.7% vs 90.0±0.7 after one-time and two-time decontamination (p<0.001). Finally, the tested FFP-3 respirator (Safe Worker 1016) had a filter capacity of 96.5%±0.7% at baseline and 60.3%±5.7% after one-time decontamination (p<0.001).

### Flow resistance

For the breathing resistance test, we tested six FFP-2 respirators (3M Aura 1862+): two were used once and reprocessed; two were used twice and reprocessed after each use; and two were used three times and reprocessed after each use. The average pressure did not increase with the number of reuses (35.6%±0.3%, 35.4%±0.0%, 36.7%±0.3%, respectively)

## DISCUSSION

The COVID-19 pandemic has caused major shortages of PPE, including protective respirators. While production has increased, shortages are so high that reprocessing of used respirators and respirators is probably one of the only viable short-term solutions. In primary care, a tabletop autoclave would be a pragmatic choice, as the device is readily available in practices for decontamination of surgical and gynaecological instruments. In this study, we found that steam decontamination at 121°C may provide a viable option for selected respirators, but it also sheds a light on the variability in the protective properties of the various available respirators and respirators. Of the tested respirators, the filter capacity of the 3M Aura 1862+ respirator fared best with a consistently high filter capacity for the 0.3 µm particle size category and above after multiple cycles of steam decontamination. Moreover, there are no indications that the respirator becomes harder to breathe through and thus more uncomfortable to wear. We also observed that with multiple decontamination cycles, the mean particle filtration efficiency for 5 microns becomes slightly lower than that for 1 micron particle for some of the respirators. We speculate that perhaps larger 5-micron particles are more likely to remain trapped in the filter material after use and during reprocessing and are subsequently sucked out of the material during testing, which in turn negatively affect the filter readings.

### Findings in relationship to FFP-2 and FFP-3 standards

The particle chamber used in this study appears to be more stringent (more sensitive) that the NEN-149 criteria that are used for FFP-2 and FFP-3 norms. We performed a cross-check with 4 KN95 respirators, which showed that measurements of 67% and 82% particle retentions at 0.3 and 0.5 microns on average, using our set-up, still

| Table 1 | Filter capacity of the tested respirators by particle size | | | | | |
|---|---|---|---|---|---|---|
| **Respirator** | **Condition (new/decontaminated)** | **No of samples** | **0.3 µm** | **0.5 µm** | **1.0 µm** | **5.0 µm** |
| 3M Aura 1862+ | New | 4 | 99.3±0.3 | 99.7±0.0 | 99.8±0.1 | 99.9±0.2 |
| | 1x | 4 | 97.0±0.8 | 99.0±0.0 | 99.0±0.5 | 100±0.0 |
| | 2x | 8 | 94.2±1.3 | 97.4±0.5 | 98.9±0.3 | 99.9±0.1 |
| | 3x | 4 | 94.4±1.3 | 97.5±0.9 | 98.8±0.4 | 100±0.0 |
| 3M Aura 9322+ | New | 2 | 96.7±0.2 | 99.1±0.0 | 99.7±0.0 | 99.3±0.3 |
| | 1x | 2 | 91.0±1.0 | 99.0±0.0 | 100±0.0 | 100±0.0 |
| | 2x | 2 | 77.5±1.5 | 85.5±0.5 | 89.5±0.5 | 98.0±1.0 |
| San Huei 2920V | New | 2 | 95.5±0.5 | 99.0±0.0 | 100±0.0 | 100±0.0 |
| | 1x | 8 | 92.3±1.6 | 97.8±0.7 | 99.1±0.3 | 96.0±5.4 |
| | 2x | 6 | 90.2±0.7 | 96.8±0.35 | 99.0±0.0 | 97.8±2.3 |
| Maco Pharma ZZM002 | 1x | 8 | 89.3±3.6 | 96.8±1.2 | 98.9±0.3 | 99.8±0.4 |
| | 2x | 6 | 90.2±0.7 | 96.8±0.35 | 99.0±0.0 | 97.8±2.3 |
| Safe Worker 1016 | New | 2 | 96.5±0.5 | 98.0±0.0 | 60.5±1.5 | 99±0.0.0 |
| | 1x | 8 | 60.3±5.3 | 81.6±4.9 | 90.1±5.3 | 91.5±18.4 |

resulted in approval for use according to the FFP-2 norm when measured according to the NEN-149, based on a continuous flow set-up (Kalibra, Delft, the Netherlands). Therefore, apart from the Safe Worker 1016, all other mask types will likely still comply with NEN-149 FFP-2 threshold values.

### Study limitations

Our study involved a pragmatic study with a limited sample size of respirators and respirators available in our practice. Our study did not involve testing of surgical masks or FFP-1 masks, and we do not know whether reprocessing of these materials would still provide adequate protection based on their respective standards. Furthermore, we did not perform a laboratory-based Fit test. Also, we presumed that solid particles of 0.3–5.0 microns are of relevance and behave similar to droplets that normally carry viruses from one person to another. Smaller particles of 0.1–0.2 microns were not included in this study, as we deemed these to contribute less to the spread of the virus. However, this is an assumption as we do not yet know for certain at what particle size viral transmission is still possible and respirators provide adequate protection.[12] Although the used flow rate of 28 L/min is in the range of the normal breathing conditions, it did not fully comply with the requirement for the EN-149 sampling flow rate. Therefore, additional studies should also include the influence on flow rate on particle filtration capacity.

### Prior studies on heat as a decontamination method

From the literature, there is a consensus that thermal inactivation is a very efficient technique to eliminate viruses.[2 6 7 11 13] Prior research indicates that steam decontamination for a total of 5 min is sufficient to completely inactivate the avian coronavirus, for instance.[14] Moreover, thermal inactivation of viruses, such as SARS-CoV-2, porcine and avian coronaviruses, poliovirus and influenza virus do not appear to differ much.[14–16] For SARS-CoV-2, a study by Fisher *et al* studied inactivation of this particular virus using four modalities, including dry heat (70°C). The study found that dry heat kills SARS-CoV-2 at a speed similar to Ultraviolet radiation.[17] Based on these combined data, it is assumed that decontamination via autoclave is also sufficient to inactivate SARS-CoV-2.

### Prior studies on the reuse of respirators

One possible concern with respirators is that extended use and reuse could reduce its protective effectiveness in terms of filter function and fit. Lin *et al* assessed the impact of steam decontamination and other decontamination procedures on the filter capacity of respirators.[18] In this study, the authors found that one of the decontamination processes that appeared effective for N95 respirators was the medical autoclave, in which they exposed the respirators to saturated steam at 121°C for 15 min. They found that filter quality (≥95%) of the masks remained intact using a range of particles. These findings are comparable to those we present in this paper. Besides filter capacity, the integrity of facepiece respirators should also be kept in mind. When exposing masks to higher temperatures (132–140°C), respirators may become deformed, as was shown in a recent study of the Dutch Centres of Disease Control (Rijksinstituut voor Volksgezondheid en Milieu (RIVM)).[19] In our study, we did not find such deformity at a lower temperature.

### Implications for practice

For COVID-19, like for other viruses, transmission can occur via droplets or aerosols.[1–4] Thus, personal protection is warranted to avoid catching COVID-19. Currently, there is no evidence on which type of face mask offers the best protection for COVID-19. Prior studies with influenza viral particles showed that FFP-2 respirators may provide better protection than surgical masks when used appropriately.[20] In this COVID-19 pandemic, it is thought that the use of surgical masks may be sufficient for consultations with only limited person-to-person exposure. However, it is much less certain whether surgical masks will provide adequate protection during longer consultations or back-to-back consultations with patients suspected to have COVID-19 in a closed consultation room.[21] In these instances, respirators may be preferable. Given the limited availability, reusing FFP-2 type respirators may provide a second-best alternative that can be readily performed in primary care and other low-resource settings using a table-top medical autoclave, as described in this study. In the unlikely event of performing or present for an aerosol-generating procedure, the Centers for Disease Control and Prevention (CDC) states that reprocessed respirators should not be used.[1] Please also be advised about the following: the use of exhalation valve-type respirators for healthcare workers is debatable. The use of an exhalation valve does not appear to offer a benefit in physiological burden over a respirator without valve for the wearer,[22] while it exposes (the often vulnerable) patient to the user's exhalation breath. As such, when available, a respirator without an exhalation valve should be preferred. We would also advice to mark reprocessed respirators with the wearer's initials, as well as the number of cycles. Finally, physicians should familiarise themselves on how to perform a user seal check. This procedure should be performed every time a respirator is put on and assures that the respirator is being properly worn. Details on how to perform this simple check can be found at the website of the CDC.[9]

### CONCLUSION

This study shows that selected FFP-2 respirators may be reprocessed for use in primary care, as the respirators retain their shape, ability to retain particles and breathing comfort after decontamination using a medical autoclave. However, future studies are warranted to confirm our findings.

**Author affiliations**

<sup>1</sup>Department of General Practice, Amsterdam UMC, Locatie AMC, Amsterdam, The Netherlands

<sup>2</sup>Department of BioMedical Engineering, Faculty of Engineering, Delft University of Technology, Delft, The Netherlands

<sup>3</sup>Van Straten Medical, De Meern, The Netherlands

<sup>4</sup>Holendrecht Medical Center, Amsterdam, The Netherlands

<sup>5</sup>Medical Centre Haaglanden, Den Haag, Zuid-Holland, The Netherlands

<sup>6</sup>GreenCycl, De Meern, The Netherlands

**Acknowledgements** The authors thank Jan-Willem Klok and Tomas Lenssen for testing all samples on the test set-up.

**Contributors** REH, BvS, BvM-vS, JB and TH contributed to the study design and acquisition of data. REH, BvS and TH analysed and interpreted the data. JB, JvdD, JvdS contributed to the interpretation. REH drafted the initial manuscript and all authors critically revised the manuscript and gave final approval.

**Funding** The authors have not declared a specific grant for this research from any funding agency in the public, commercial or not-for-profit sectors.

**Competing interests** BvS has shares in the company Van Straten Medical and Greencycle that builds and owns a part of the measurement systems used for the execution of this study. Greencycle facilitated the measurements.

**Patient consent for publication** Not required.

**Provenance and peer review** Not commissioned; externally peer reviewed.

**Data availability statement** Data are available in a public, open access repository. All data will be made available online (https://repository.tudelft.nl/).

**ORCID iD**

Ralf E Harskamp http://orcid.org/0000-0001-9041-0350

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
