## [Reviewer comments · BMJ Open]

ARTICLE DETAILS

TITLE (PROVISIONAL)	Reprocessing filtering facepiece respirators in primary care using medical autoclave: a prospective, bench-to-bedside, single-center study
AUTHORS	Harskamp, Ralf; van Straten, Bart; Bouman, Jonathan; van Maltha - van Santvoort, Bernadette; van den Dobbelsesteen, John; van der Sijp, Joost; Horeman, Tim

VERSION 1 – REVIEW

REVIEWER	Chane-Yu Lai Chung Shan Medical University, Taichung, Taiwan
REVIEW RETURNED	25-May-2020

GENERAL COMMENTS	1. The introduction or discussions could compare or cite more references on decontamination of N95FFRs, especially Raymond J. Roberge, Ronald E. Shaffer and Chane-Yu Lai.2. In methods, the fit test is more useful than user seal check.3. Also, in particle penetration test, “the 28 liter of surrounding air is sucked through the filter material of the mask and the remaining particles are counted in the categories of 0.3, 0.5, 1.0 and 5.0microns”. The 28 liter was not enough or match the requirement for the EN-149 sampling flow rate or face velocity. Moreover, the tested particle size barely included the most penetrating particle size (MPPS) of electret N95FFRs.4. Masks that can reuse, not only including fit and decreased risk of spreading other diseases, but also the filter quality or acceptable filtration efficiency, and bioefficacy. Suggest add some references or redo some experiments.5. Strongly recommend the authors to calculate the filter quality of each tests before and after decontamination, and describe the discrepancies between the decontamination.6. The whole article did not compare the bioefficacy of any bacteria or viruses after decontamination. Did the decontamination process work well or not? What are the decontamination criteria? How do the authors make sure that the decontamination method is good or not?
---

REVIEWER	David Ozog Henry Ford Hospital USA
REVIEW RETURNED	30-May-2020

GENERAL COMMENTS	Thank you for this interesting work during this challenging time. In the manuscript you alternate between decontaminate and sterilization. Decontamination is the accepted terminology worldwide - please remove sterilization.
---

	Please comment on dry vs moist heat. Autoclave devices are variable in setting/temperature/time - the results on your device may not translate. Dry/Moist heat has been done for COVID-19 and 70C for 30 minutes did not inactivate Fisher Morris et al 2020. This may partially support your higher temp/time/humidity. However, a single H1N1 test is not an appropriate surrogate - please comment. By the time this is published, the few cases of anecdotal uses of cloth masks will be dated. It has been recommended that health care workers do not use masks with exhalation valves. Please comment. The 3M 1860 series of respirators has some of the highest structural integrity. The challenge during the shortage is that the "best" respirators are not available and many of the others would potentially fail fit testing. Please comment on the importance of fit-testing ANY respirator. Ozog et al JAAD 2020. Any visible contamination including saliva/mucous/external fluid will affect results. Subjects should only reuse their own respirators in this setting. The process is to label respirators as has been done is hydrogen peroxide FDA EUA. Decontaminated respirators should not be used for aerosolizing procedures, regardless of decontamination method. Please add. Thank you for this work which can potentially benefit small offices that currently have autoclave devices, understand fit testing, and have access to high quality respirators. The limited capacity (4 respirators over 34 minutes) precludes widespread use.
--	---

REVIEWER	Wen-Yinn Lin National Taipei University of Technology Taiwan
REVIEW RETURNED	03-Jun-2020

GENERAL COMMENTS	 1. Particle generation system and number concentration of different sizes are suggested to be provided. 2. What could be the reason why the filter capacities of one- and two-time sterilized San Huei 2920V for 5.0 micrometer-sized particles were lower than that for 1.0 micrometer. 3. The filter capacities of new Safe Worker 1016 (1.0 micrometer) and one-time sterilized (0.3 micrometer) were significantly lower than other particle sizes. Moreover, the capacity for 0.3 micrometer decreased from 96.5 to 60.3 after one-time sterilization, while it (for 1.0 micrometer) increased from 60.5 to 90.1. What could be the reason? 4. Did deformity result in significant change of flow resistance?
---

VERSION 1 – AUTHOR RESPONSE

Reviewer: 1

Comment 1. The introduction or discussions could compare or cite more references on decontamination of N95FFRs, especially Raymond J. Roberge, Ronald E. Shaffer and Chane-Yu Lai.
Response 1. Dr Roberge is mentioned in the discussion on the physiological impact of valve-type and

non-valved N96 filtering facepiece respirators. Dr. Shaffer is mentioned in reference 9 regarding the user seal check. We put information on this procedure in the discussion part of the manuscript. Dr. Chane-Yu Lai is also included in the manuscript, namely cited in the important paper on (PloS ONE 2017;12:e0186217).

Comment 2. In methods, the fit test is more useful than user seal check.

Response 2. While this is correct; the user seal check is test that can be easily performed in the clinic without the need of scarcely available and expensive FIT test equipment and should therefore also be included.

Comment 3. Also, in particle penetration test, “the 28 liter of surrounding air is sucked through the filter material of the mask and the remaining particles are counted in the categories of 0.3, 0.5, 1.0 and 5.0microns”. The 28 liter was not enough or match the requirement for the EN-149 sampling flow rate or face velocity. Moreover, the tested particle size barely included the most penetrating particle size (MPPS) of electret N95FFRs.

Response 3. As these suggestions are very valid, we now stated in the limitations: *“Also, we presumed that solid particles of 0.3-5 microns are of relevance and behave similar as droplets that normally carry viruses from one person to another. Smaller particles of 0.1-0.2 microns were not included in this study, as we deemed these to contribute less to the spread of the virus. However, this is an assumption as we do not yet know for certain at what particle size viral transmission is still possible and respirators provide adequate protection. [12] Although the used flow rate of 28 liter/min is in the range of the normal breathing conditions it did not fully comply with the requirement for the EN-149 sampling flow rate. Therefore, additional studies should also include the influence on flow rate on particle filtration capacity.”*

Comment 4. Masks that can reuse, not only including fit and decreased risk of spreading other diseases, but also the filter quality or acceptable filtration efficiency, and bioefficacy. Suggest add some references or redo some experiments.

Response 4. We have rewritten relevant sections in the discussion paragraphs. Please see the paragraphs “prior studies on heat as a decontamination method” and “prior studies on the reuse of respirators”.

Comment 5. Strongly recommend the authors to calculate the filter quality of each tests before and after decontamination, and describe the discrepancies between the decontamination.

Response 5. We have calculated the filter quality of each sample, of unused respirators, as well as single, and multiple decontamination cycles. Please see *table 1*.

Comment 6. The whole article did not compare the bioefficacy of any bacteria or viruses after decontamination. Did the decontamination process work well or not? What are the decontamination criteria? How do the authors make sure that the decontamination method is good or not?

Response 6. Our group previously demonstrated that steam decontaminated respirators proved negative for bacteria [11]. Others have performed further study on bioefficacy. We have added the following paragraph to the discussion section of the manuscript. (please see italic text below). Moreover in our “implications for practice” paragraph we have also provided practical tips, such as: 1) do not use reprocessed respirators in the event of aerosol-generating procedures (per CDC recommendations); 2) mark reprocessed respirators with the wearer’s initials as well as the number of reprocessing cycles.

“From the literature there is a consensus that thermal inactivation is a very efficient technique to eliminate viruses. [2,6,7,11,13] Prior research indicates that steam decontamination for a total of 5 minutes is sufficient to completely inactivate the avian coronavirus, for instance [16]. Moreover, thermal inactivation of viruses, such as SARS-CoV, porcine and avian coronaviruses, poliovirus, and influenza virus do not appear to differ much. [14-16] For SARS-CoV-2, a study by Fisher et al studied inactivation of this particular virus using four modalities, including dry heat (70 degrees Celsius). The

study found that dry heat kills SARS-CoV-2 at similar speed to UV. [17] Based on these combined data it is assumed that decontamination via autoclave is also sufficient to inactivate SARS-CoV-2."

Reviewer: 2

Comment 1. In the manuscript you alternate between decontaminate and sterilization. Decontamination is the accepted terminology worldwide - please remove sterilization.

Response 1. We agree with the reviewer and have changed "sterilization" to "decontamination"

Comment 2. Please comment on dry vs moist heat. Autoclave devices are variable in setting/temperature/time - the results on your device may not translate. Dry/Moist heat has been done for COVID-19 and 70C for 30 minutes did not inactivate Fisher Morris et al 2020. This may partially support your higher temp/time/humidity. However, a single H1N1 test is not an appropriate surrogate - please comment.

Response 2. We have added a paragraph in the discussion section of the manuscript on the use of heat for decontamination, in which we also included the findings of Fisher et al.

Comment 3. By the time this is published, the few cases of anecdotal uses of cloth masks will be dated.

Response 3. We have slightly rephrased this sentence.

Comment 4. It has been recommended that health care workers do not use masks with exhalation valves. Please comment.

Response 4. We have added information on this topic in the paragraph "implications for practice", which can be found in the discussion section of the manuscript.

Comment 5. The 3M 1860 series of respirators has some of the highest structural integrity. The challenge during the shortage is that the "best" respirators are not available and many of the others would potentially fail fit testing. Please comment on the importance of fit-testing ANY respirator. Ozog et al JAAD 2020.

Response 5. We agree that the 3M 1860 series provide high structural integrity. While we agree that fit and material testing is of at most importance as one cannot determine the quality of a respirator in any other way, it is our belief that this is somewhat out of the scope of this manuscript and the message we bring to the audience.

Comment 6. Any visible contamination including saliva/mucous/external fluid will affect results. Subjects should only reuse their own respirators in this setting. The process is to label respirators as has been done is hydrogen peroxide FDA EUA.

Response 6. We agree that this will affect the results.

Comment 7. Decontaminated respirators should not be used for aerosolizing procedures, regardless of decontamination method. Please add.

Response 7. We agree that this is the case, however in primary care settings these types of procedures are extremely rare.

Reviewer: 3

Comment 1. Particle generation system and number concentration of different sizes are suggested to be provided.

Response 1. We thank the reviewer for this comment. We have revised the paragraph “particle penetration test” to further clarify the procedure we followed, as well as updated the figure. Textual changes include the following: *“Before each measurement, a benchmark test is conducted with 28 Liters of surrounding air that is sucked through the particle chamber into the particle counter. The particle counter measures the particles that are naturally present in the air.”*

Comment 2. What could be the reason why the filter capacities of one- and two-time sterilized San Huei 2920V for 5.0 micrometer-sized particles were lower than that for 1.0 micrometer.

Response 2. To explain this phenomenal we now added to the discussion: *“We also observed that with multiple decontamination cycles , the mean particle filtration efficiency for 5 microns becomes slightly lower than for 1 micron particles for some of the respirators. We speculate that perhaps larger 5 micron particles are more likely to remain trapped in the filter material after use and during reprocessing and are subsequently sucked out of the material during testing, which in turn negatively affect the filter readings.”*

Comment 3. The filter capacities of new Safe Worker 1016 (1.0 micrometer) and one-time sterilized (0.3 micrometer) were significantly lower than other particle sizes. Moreover, the capacity for 0.3 micrometer decreased from 96.5 to 60.3 after one-time sterilization, while it (for 1.0 micrometer) increased from 60.5 to 90.1. What could be the reason?

Response 3. We attribute this effect to treatment with the medical autoclave which negatively affects the integrity of the respirator filter material.

Comment 4. Did deformity result in significant change of flow resistance?

Response 4. The respirators with significant deformity were not tested for change in flow resistance, as these were deemed not acceptable for use in clinical practice.

VERSION 2 – REVIEW

REVIEWER	David Ozog Henry Ford Hospital, USA
REVIEW RETURNED	15-Jul-2020
GENERAL COMMENTS	Thank you for the updates. The manuscript is improved.